# The Effect of Heat Stress on Wheat Flag Leaves Revealed by Metabolome and Transcriptome Analyses During the Reproductive Stage

**DOI:** 10.3390/ijms26041468

**Published:** 2025-02-10

**Authors:** Shuonan Duan, Xiangzhao Meng, Huaning Zhang, Xiaotong Wang, Xu Kang, Zihui Liu, Zhenyu Ma, Guoliang Li, Xiulin Guo

**Affiliations:** Institute of Biotechnology and Food Science, Hebei Academy of Agriculture and Forestry Sciences/Hebei Key Laboratory of Plant Genetic Engineering, Shijiazhuang 050051, China; duanshuonan@haafs.org (S.D.); meng.xiang.5@163.com (X.M.); zhn.8888-@163.com (H.Z.); wxt15230091065@163.com (X.W.); kangx1998@163.com (X.K.); liuzihui1978@163.com (Z.L.); mazhenyuqqtt@163.com (Z.M.)

**Keywords:** wheat, heat stress, flag leaf, amino acids, HSP, ABC transporter

## Abstract

In this study, we were dedicated to investigating the effect caused by heat stress on wheat flag leaves. Metabolome and transcriptome analysis were introduced to identify some key biological processes. As a result, 182 and 214 metabolites were significantly changed at the anthesis and post-anthesis stages, respectively; most of them were lipids, amino acids and derivatives, phenolic acids, and alkaloids. Aminoacyl-tRNA biosynthesis was the most significantly enriched pathway by metabolites at both two stages, each of which included 13 types of amino acid, and 12 of them were shared and up-regulated. Therefore, we further measured 22 kinds of amino acid content in ten different wheat genotypes at the post-anthesis stage. Based on the average content of each amino acid, 17 kinds of them were significantly increased after heat stress, and 4 types were significantly decreased. Both the metabolism analysis and the transcriptome analysis had a higher number of significantly changed metabolites or differential expressed genes at the post-anthesis stage, which indicated that the post-anthesis stage is more sensitive to heat stress, with 21,361 and 17,130 differential expressed genes, respectively. Two pathways, protein processing in endoplasmic reticulum and ABC transporters, were significantly enriched at two stages. The differential expressed genes in processing in endoplasmic reticulum pathway mainly encoded various types of molecular chaperones; among them, the HSP20 family was the most predominant and intensively up-regulated. The ABC transporter gene family is another pathway that is deeply involved in heat-stress response, which could be classified into five subfamilies; among them, subfamilies B and G were the most active. In summary, this study revealed the heat response pattern of amino acids, HSPs, and ABC transporter which may play a vital role during the wheat reproductive stage.

## 1. Introduction

The diploid progenitors of wheat were adapted to the dry, cool-summer Mediterranean climate initially, which made wheat relatively sensitive to high temperatures [1]. The complex genome compositions driven by polyploidization events provide wheat with greater physiological and ecological plasticity than its diploid progenitors, and wide adaptability to harsh growth environments, which has enabled wheat to cultivate well beyond the ranges of its progenitors and become a global staple food crop [2]. With the increasing demand for wheat, wheat is grown in tropical and sub-tropical regions of the world, which leads wheat to encounter heat stress more intensively and frequently.

Heat stress poses a threat to a wide range of plant metabolic pathways, including membrane stability, photosynthesis and starch synthesis [3]; among them, starch synthesis is highly sensitive to heat stress. As a result, the reduction in starch accumulation can exceed 30% at temperatures between 30 °C and 40 °C in wheat grains [4]. The optimal daytime growing temperature of wheat is 21 °C during reproductive development; every degree Celsius above this optimum leads to a reduction in yield of 3–4% [5,6]. Agronomic traits are affected by heat stress at every wheat developmental stage [4,7]. Heat stress results in reduced grain number per spike and ability of pollen germination in the anthesis stage, yield losses by abortion of grains, decreased grain weight during the post-anthesis stage, and adversely affects grain quality during the grain filling stages [8].

Wheat flag leaves are considered to be the main organ for photosynthesis and contribute 45–58% of photosynthetic performance during the grain-filling stage. However, wheat flag leaves are particularly vulnerable to heat stress, which significantly depresses flag leaf photosynthesis, enhanced cell-membrane peroxidation, reactive oxygen accumulation, and activities of antioxidative enzymes during the reproductive stage [4,9]. Feng et al. compared the heat stress influence on two wheat cultivars’ flag leaves with different heat resistance levels (low/high sensitive) during the reproductive stage. The result revealed that the highly sensitive cultivar suffered deeper impacts on photosynthetic characteristics. In addition, the low-sensitive cultivar showed some self-repair capacity after heat stress [10].

To adapt to stressful environments, terrestrial plants trigger a wide range of metabolites and genes participating in abiotic stress response, some of which swiftly alter and confer stress resistance. Plants encounter heat stress, which is always accompanied by drought stress, as the evapotranspiration leading to drought stress could be increased by heat stress [11]. For example, all aerial organs of land plants are covered with a hydrophobic cuticle layer that protects against biotic and abiotic stresses [12]. ABC protein as a powerful transporter could drive the compound exchange across different biological membranes using the energy released from ATP hydrolysis and is shown to transport hydrophobic coating materials to protect plants from drought stress damage [13]. So, the role of ABC transporters in heat stress is worth investigating.

Plants have developed various strategies to minimize the adverse effects caused by abiotic stress environments and several of strategies are related to amino acid metabolism. Accumulation of free amino acids and secondary amino acids derives under stress conditions has been reported repeatedly, which can be caused by stress-induced protein breakdown and synthesis of particular amino acids with specific beneficial functions [14,15,16]. Proline is one of the most extensively reported amino acids, which is positively associated with stress tolerance [17,18]. Proline not only regulates cell-membrane stability and balances osmotic pressure under different stress environments but also regulates the redox balance and energy status and acts as a signaling molecule to modulate specific aspects of the stress response and development of plants [19,20]. BCAAs including isoleucine, leucine, and valine have been extensively reported to be accumulated in plants in response to drought stress [14,21]. BCAAs can act as compatible osmolytes or alternative energy sources to help or enhance plant drought tolerance [22,23]. Heat stress and drought stress normally occur simultaneously in field, however, the function of BCAAs and other amino acids in response to heat stress has not been investigated in plants.

The integrated metabolic and transcriptomic analysis has been applied in dissecting molecular mechanisms of stress response mechanism and investigations of many stresses in crops successfully and systematically. Glaubitz et al. integrated transcriptomic and metabolomics analysis of six rice cultivars with different heat-tolerant levels. Some genes and pathways were identified and specific to tolerant or sensitive cultivars [24]. Boehlein et al. compared maize transcriptome profiles and metabolism profiles at elevated day or night temperatures, and the results revealed that the normal developmental program of endosperm is fully executed under prolonged high-temperature conditions, but at a faster rate, which leads to a reduction in grain dry weight [25]. Obata et al. investigated transcriptome and metabolism profiles of ten maize cultivars with different stress tolerance under heat stress or drought stress, and the results revealed that drought stress evoked the accumulation of many kinds of amino acids, and the combination of drought and heat evoked most of the metabolic changes were the sum of the responses to individual stresses. Glycine and myoinositol metabolites identified were significantly correlated with grain yield under drought [26]. Above all, the integrated metabolic and transcriptomic analysis had been proven to be a powerful tool as the metabolic trait was the connection between gene and phenotype traits which makes a more direct effect on phenotype traits [27]. Moreover, knowledge about the metabolic profile is useful for understanding the physiological traits of wheat and other crops and can help improve trait quality and stress resistance.

Regarding global warming, wheat encounters an increasing frequency of heat stress worldwide, and it will significantly adversely affect grain setting, duration, rate, quality, and finally grain yield. To minimize the effect of heat stress, molecular studies of such metabolites play a crucial role in knowing the mechanism underlying stress tolerance. In this study, we treated wheat plants removed from the field during the anthesis and post-anthesis stages with short-term heat stress in an incubator; then, the flag leaves were collected for metabolic and transcriptomic analysis. We aim to explore new insights into wheat responses to heat stress and dissect novel approaches including some metabolites and genes that were closely related to wheat heat resistance which could be used to mitigate the impacts of heat stress occurring at the reproductive stages.

## 2. Result

### 2.1. Metabolite Profiles Indicate That Pathways Related to Amino Acids Were the Most Active by Heat Stress

Metabolite profiles of CK1 (Control Check at anthesis stage), HS1 (Heat stress at anthesis stage), CK7 (Control Check at 7 days post-anthesis), and HS7 (Heat stress at 7 days post-anthesis) were performed by UPLC-MS/MS. The qualitative and quantitative metabolites were based on matching the retention time, count per second, and area of the peak to the Metware database. Totally, 1081 metabolites were detected by this study, and most of them were flavonoids (289), phenolic acids (170), lipids (139), alkaloids (99), amino acids and derivatives (93), organic acids (65), nucleotides and derivatives (63), lignans and coumarins (37), and terpenoids (17) (Appendix A). Based on the species and relative content of metabolites, the unsupervised PCA was performed to illustrate heat stress impact on the global metabolism of the two comparison groups (Figure 1). This analysis included four sample groups; each sample group comprised three replicates. The PCA analysis revealed that replicated samples from the same group were gathered closely due to the high consistency among replicates. The first two principal components (PC1 and PC2) explained 31.75% and 19.75% of the variation, respectively. PC1 discriminated the samples at the anthesis stage from the samples at the post-anthesis stage, and PC2 discriminated CK from HS (Figure 1). The replicates of CK1 and replicates of HS1 showed a relatively bigger separation compared to replicates of CK7 and HS7, which indicated a more extensive impact on metabolism at the anthesis stage by heat stress.

Among 1081 detected metabolites, 298 kinds of nonredundant metabolites were changed significantly (fold change ≥ 1.5 or fold change ≤ 0.67, VIP ≥ 1) by short-term heat stress, and 182 and 214 kinds of significantly changed metabolites were detected at two stages. Among them, 104 and 92 metabolites were significantly up-regulated, and 78 and 122 metabolites were significantly down-regulated (Appendix A). The significantly changed metabolites mainly consisted of lipids (54), amino acids and derivatives (49), phenolic acids (48), alkaloids (46), nucleotides and derivatives (25), organic acids (23), flavonoids (20), lignans and coumarins (6), and terpenoids (2) (Figure 2). The details of significantly changed metabolites in each comparison group are listed in Appendix A. Lipids, amino acids and derivatives, phenolic acids, and alkaloids were the predominant significantly changed metabolites in both comparison groups. Most of the lipids were down-regulated in CK7 vs. HS7 (31/36). Most amino acids and derivatives were up-regulated in both CK1 vs. HS1 (25/36) and CK7 vs. HS7 (28/39). Most of phenolic acids were down-regulated in CK7 vs. HS7 (26/35). Most of the alkaloids were up-regulated in both CK1 vs. HS1 (26/39) and CK7 vs. HS7 (22/33). In general, amino acids and derivatives, and alkaloids had a similar changing pattern in response to heat stress at two stages, which indicated these up-regulated amino acids and derivatives and alkaloids may play an important role in heat-stress response (Appendix A).

We analyzed the significantly changed metabolites with consistently up- or down-regulated at two stages (Figure 3). Fifty-three metabolites were consistently up-regulated by heat stress at two stages, most of them were alkaloids (18/53) and amino acids and derivatives (21/53). The consistently up-regulated alkaloids included indole, indole-3-carboxaldehyde, 3-indole acetonitrile, 3-indole acrylic acid, 3-indole propionic acid, 1-methoxy-indole-3-acetamide, methoxyindoleacetic acid, petrolatum, 4-hydroxymandelonitrile, and 1-methoxy-indole-3-acetamide, which participate in tryptophan metabolism pathway. The up-regulated amino acids and derivatives at two stages included proline, valine, threonine, cycloleucine, leucine, isoleucine, glutamine, lysine, methionine, homomethionine, phenylalanine, arginine, tyrosine, and tryptophan. In addition, three lipid metabolites were also up-regulated at two stages including palmitoleic acid, adipic acid, and homogentisic acid. Moreover, 24 metabolites were down-regulated by heat stress at both stages including 6-methoxy-2-benzoxazolinone, homoarginine and L-leucyl-L-leucine, pinobanksin, syringaresinol, lysoPE, argininosuccinic acid, chlorogenic acid which belong to alkaloids, amino acids and derivatives, flavonoids, lignans and coumarins, lipids, organic acids, and phenolic acids.

The significantly changed metabolites were taken into the metabolite pathway enrichment analysis to investigate and compare heat-stress responses in wheat flag leaves during the reproductive stage. There were 13 and 14 metabolite pathways significantly enriched in CK1 vs. HS1 and CK7 vs. HS7, respectively, by KEGG analysis (*p* < 0.05) (Appendix A). Eight of these pathways were shared at two stages including aminoacyl-tRNA biosynthesis, glucosinolate biosynthesis, biosynthesis of amino acids, 2-oxocarboxylic acid metabolism, D-amino acid metabolism, ABC transporters, valine, leucine and isoleucine biosynthesis, and cyanoamino acid metabolism. The specific enriched pathways of each stage consisted of five and six pathways in CK1 vs. HS1 and CK7 vs. HS7, respectively, such as glycine, serine and threonine metabolism; monobactam biosynthesis; valine, leucine, and isoleucine degradation; and tryptophan metabolism were specifically enriched in CK1 vs. HS1; and phenylalanine; tyrosine and tryptophan biosynthesis; cyano amino acid metabolism; pyruvate metabolism; starch and sucrose metabolism; and phenylalanine metabolism were specifically enriched in CK7 vs. HS7 (Figure 4). Aminoacyl-tRNA biosynthesis was the most significantly enriched pathway in both stages; each of these pathways included 13 types of amino acids, among which 12 were shared and all of the amino acids in this pathway were up-regulated by heat stress except L-serine. In addition, some metabolite pathways were enriched at two stages due to the increased amino acids, such as biosynthesis of amino acids, D-amino acid metabolism, ABC transporters, valine, leucine, and isoleucine biosynthesis. In general, we detected several amino acids and their derivatives that could be responsive to heat stress via various pathways.

### 2.2. The Amino Acids Changed by Heat Stress in Ten Wheat Cultivars’ Leaves

As amino acids were the most active metabolites detected in metabolism analysis, the flag leaves of ten wheat cultivars were used to measure the amino acids’ heat-stress response pattern. Among 22 amino acids detected in this study, the contents of homocitrulline, cystathionine, homoarginine, and methionine were at a very low level either under control or heat-stress conditions. The highest content was observed in glutamine, the average content of which was more than 6000 μg/g under both control and heat-stress conditions, aspartate, serine, and alanine also had a high abundance. The heat stress led seventeen amino acids to change significantly in the reproductive stage leaves, most of which were up-regulated by heat stress (*p* < 0.05). There were fourteen significantly up-regulated amino acids including valine, tyrosine, tryptophan, isoleucine, histidine, arginine, alanine, threonine, lysine, and leucine, besides four kinds of amino acids that were down-regulated significantly by heat stress. In general, certain amino acids showed a similar change pattern among ten different wheat cultivars (Appendix A).

### 2.3. DEGs Encoding Molecular Chaperone Were the Most Active to Heat Stress

The samples and comparison groups used in metabolic analysis were also used for transcriptome analysis. The data generated from the global transcript was used in PCA analysis, and the result revealed that the replicates in the same group were gathered closely, whereas the different sample groups were separated clearly which indicates the transcriptional reprogram represented both plant development and heat stress (Figure 5). There were 17,130 and 21,361 DEGs detected in CK1 vs. HS1 and CK7 vs. HS7, respectively, which consisted of 9418 and 11,058 up-regulated genes, 7712 and 10,303 down-regulated genes, and there were 6429 up-regulated genes and 3936 down-regulated genes overlapped (Appendix A). To identify the main pathways and functional genes involved in heat-stress response in wheat flag leaves, the GO and KEGG analyses were performed, and both approaches were used to uncover metabolism modified by heat stress. There were 45 and 37 significantly enriched pathways in CK1 vs. HS1 and CK7 vs. HS7 by KEGG analysis, respectively (*p* < 0.05), and 22 pathways were shared including protein processing in endoplasmic reticulum; glutathione metabolism; ABC transporters; valine, leucine, and isoleucine degradation (Appendix A). Among them, the protein processing in endoplasmic reticulum was the most significantly enriched pathway at two stages (Figure 6), which enriched 309 and 396 DEGs, respectively, most of which served as molecular chaperones (Appendix A). Among them, 259 DEGs were up-regulated at both stages, and 12 DEGs were down-regulated after heat stress. Most of the up-regulated DEGs were HSP family members (160/259), including HSP20 (106/259), HSP70 (35/259), and HSP90 (19/259) family proteins. In addition, there were also some up-regulated DEGs during the reproductive stage encoding calnexin, DnaJ, HSP70-interacting protein, and protein disulfide-isomerase. In this study, a higher number of DEGs were detected at the post-anthesis stage than the anthesis stage from this pathway, which meant some DEGs were specifically expressed at the post-anthesis stage. We identified 125 DEGs that were only significantly expressed at the post-anthesis stage encoding ubiquitin-conjugating enzyme, protein transport protein, and ubiquitin fusion degradation protein (Appendix A).

GO-term analysis was performed to identify heat stress-responsive gene terms in cellular component, biological process, and molecular function ontology sections (Appendix A). The GO-term analysis of CK1 vs. HS1 and CK7 vs. HS7 identified 10 and 23 significantly enriched GO-terms in cellular components, 301 and 250 terms in biological process, and 147 and 136 terms in molecular function (*p* < 0.05). In biological process ontology, 122 GO terms were shared; among them, the response to the hydrogen peroxide term was the most significantly enriched at two stages, and eleven GO terms were related to protein folding. In addition, there were 15 GO terms related to metabolite transportation including amino acid transport, anion transmembrane transport, lipid translocation, phospholipid transport, carbohydrate derivative transport, sodium ion transport, organic cation transport, and calcium ion transport. Some GO terms closely related to plant stress response were also significantly enriched and shared, such as phytohormone metabolism, oxidative stress, cellular response to heat, and heat acclimation. In line with biological process ontology, the shared GO terms in molecular function ontology were mainly related to protein binding and transporter activity including unfolded protein binding, chaperone binding, heat shock protein binding, misfolded protein binding, and organic acid transmembrane transporter activity, amino acid transmembrane transporter activity, water transmembrane transporter activity, neutral amino acid transmembrane transporter activity, arginine transmembrane transporter activity, some sugar and alcohols metabolite transmembrane transporter activity. In general, the transcriptome rearrangement of short-term heat stress is mainly related to protein binding and metabolite transportation.

### 2.4. Integrated Transcriptome and Metabolome Analysis

The data generated from the transcriptome and metabolome analysis was integrated to reveal the impacts caused by heat stress on wheat flag leaves during the reproductive stages. Although a wide-targeted metabolome method was introduced in this study, a limited number of metabolites were qualified and quantified. However, some pathways were selected in this study, which were enriched by both significantly changed metabolites and DEGs to provide deeper insight into the heat-stress response in wheat flag leaves. Biosynthesis of amino acids; ABC transporters; glycine, serine, and threonine metabolism; cyano amino acid metabolism; valine, leucine, and isoleucine degradation; and metabolic pathways were both significantly enriched in transcriptome and metabolome analysis at the anthesis stage. ABC transporters, and starch and sucrose metabolism were both significantly enriched in transcriptome and metabolome analysis at the post-anthesis stage (Appendix A). In this study, we focused on ABC transporter and biosynthesis of amino acids, because these two pathways contained a higher number of significantly changed metabolites and DEGs caused by heat stress.

ABC transporter was the only pathway significantly enriched in integrated analysis at two stages (Appendix A), as eighteen and twenty-one metabolites, 116 and 122 DEGs were detected in CK1 vs. HS1 and CK7 vs. HS7, respectively. Thirteen metabolites overlapped with the consistent heat stress change pattern including 5-aminolevulinic acid, inositol, S-methyl-L-cysteine, trans-4-hydroxy-L-proline, and nine kinds of amino acids (Appendix A). All DEGs in this pathway were ABC transporter family members belonging to subfamilies A, B, C, D, and G (Appendix A). The expression pattern of ABC transporter family members revealed a relatively higher number of down-regulated genes than up-regulated genes at both stages, and there was no consistent expression pattern of ABC transporter family members in each subfamily.

The biosynthesis of amino acids pathway was significantly enriched in CK1 vs. HS1 and CK7 vs. HS7 of metabolism analysis and CK1 vs. HS1 transcriptome analysis. Although the CK7 vs. HS7 of transcriptome analysis was not significantly enriched, many DEGs were found in the biosynthesis of amino acids pathway. The number of metabolites found in the biosynthesis of amino acids pathway of CK1 vs. HS1 and CK7 vs. HS7 were 19 and 22, respectively, most of which were increased after heat stress (Appendix A). The number of DEGs found in the biosynthesis of amino acids pathway of CK1 vs. HS1 and CK7 vs. HS7 were 195 and 173, respectively, coding dehydrogenase, synthase, transferase, transketolase, transaminase, aldolase, enolase, isomerase, hydrolase, kinase, which were specifically related to alanine, aspartate, glutamate, valine, leucine isoleucine, arginine, and proline metabolism (Appendix A).

### 2.5. The Verification of Transcriptome Data by RT-qPCR

To verify the transcript level, the expression pattern of selected genes including five molecular chaperones (Figure 7A–E) and three ABC transporters (Figure 7F–H) was measured by RT-qPCR. The expression of molecular chaperones was significantly increased at two stages, especially the HSPs which had the biggest up-regulated fold change after heat stress. The expression level of ABC transporters (ABCB1 and ABCB19) was decreased by heat stress significantly. The relative expression levels revealed by RT-qPCR were consistent with the FPKM values from the transcriptomic data, which indicated a relatively high consistency.

## 3. Discussion

Heat stress could have adverse effects on agronomic traits of wheat at all production stages, and the reproductive stage is relatively more sensitive to heat stress [28]. In addition, short-term heat stress can reduce grain number per spike and yield at anthesis stages, which can be attributed to the lower ability of pollen to germinate, and to the rate of pollen tube growth [29]. During the anthesis stage, yield losses were caused by the abortion of grains and reduced grain weight [30]. As large-scale metabolism and transcriptome analysis of flag leaves under heat stress in the field remain rare, and the underlying mechanisms remain unclear, we performed wide-targeted metabolome and transcriptome analysis to explore and compare the responses of heat stress in flag leaves during the reproductive stage to uncover the pathways and metabolites that may be closely related to heat stress resistance or tolerance.

### 3.1. Various Molecular Chaperones Were Activated by Heat Stress During the Reproductive Stage

Heat stress could disturb protein folding in the endoplasmic reticulum, which has a sophisticated quality system as endoplasmic reticulum quality control. This process needs to up-regulate genes encoding molecular chaperones for increasing protein-folding capacity, and most of the molecular chaperones were originally identified as HSPs [31]. Based on the annotation, we identified that most of the DEGs encoded chaperones in protein processing in endoplasmic reticulum pathway, and these DEGs were classified into fifteen groups. The HSP20 group was the most active gene family that responded to heat stress during the reproductive stage, as the HSP20 group identified in this study contained the highest number of DEGs and was also the most strongly up-regulated group compared to other groups (Figure 8). HSP20 was referred to as sHSP, which could rather than refold non-native proteins by themselves, bind non-native proteins and stabilize and prevent non-native aggregation, thereby facilitating their subsequent refolding by ATP-dependent chaperones [32,33]. The sHSPs are the most prevalent in plants with several families [34]. The abundant diversity of sHSP was associated with a wide range of environmental stress adaption [35,36]. The second biggest group of this pathway in this study was the HSP70, which could assist refolding of non-native proteins under both normal and stress conditions [37,38]. Some HSP70s are constitutively expressed, which are often involved in assisting the folding of de novo synthesized. Other family members are only activated after stress. Therefore, they are more involved in non-native protein refolding and proteolytic degradation [37,38,39]. Among the significantly changed HSP70s of this study, most of them were up-regulated during the reproductive stage, which indicated that these HSP70s could respond to heat stress (Figure 8). In addition, HSP70s never function alone. They invariably require a J protein as cofactors. These cofactors are keys, as they regulate the binding of HSP70s to client proteins by affecting HSP70s’ interaction [40]. In this pathway, we also identified some DEGs encoding J protein significantly up-regulated including DnaJs, which are referred to as HSP40s, and HSP70-interacting proteins. Much of the functional diversity of the HSP70s is determined by a diverse class of cofactors: J proteins, therefore we believed that the DEGs encoding the DnaJ, HSP70-interacting protein, and HSP70s in this study may interact and specifically respond to heat stress in flag leaves during the reproductive stage. HSP90s as another kind of chaperone were also considered to play a vital role in plant heat-stress response. The best known of HSP90s was its activation of signaling proteins such as protein kinases, hormone receptors, and transcription factors in eukaryotic cells [41]. HSP90s and HSP70s could interact and negatively regulate the heat stress transcription factors, which are essential for heat-stress responses through activation of the series of heat-stress genes [42,43]. In this study, we figured out some HSP90s that were involved in heat-stress response, and most of them were significantly up-regulated. In addition, a wide range of chaperones were identified in this study, most of which were significantly up-regulated including calnexin, calreticulin, protein disulfide-isomerase, and mannosidase (Figure 8). Above all, there was a strong activation of various molecular chaperones caused by heat stress, which indicated that a strong demand for protein-folding capacity was triggered by heat stress. Previous research showed that HSPs could be induced by plant alkaloids, consequently enhancing the heat tolerance of *Arabidopsis* [44]. In this study, we also identified 26 and 22 types of alkaloids that were significantly up-regulated under HS at the two stages, and 18 alkaloids were shared, including 1-Methoxy-indole-3-acetamide and indole, which were intensively introduced by heat stress.

In conclusion, this study could provide further insight into the mechanism that how wheat copes with the increasing misfolded protein made by heat stress during the reproductive stage.

### 3.2. The Roles of ABC Transporter Subfamilies in Response to Heat Stress

ABC transporters drive the exchange of compounds across many different biological membranes, using the energy released from ATP hydrolysis [45]. The materials transported by ABC transporters include surface coating materials, defense molecules, supportive materials, secondary metabolites, and plant hormones that regulate the overall development of plants. The ABC transporter has been proven that it is related to the plant’s adaption to stressful environments by transporting materials for surface coating [12]. In this study, the wheat plants were treated with short-term heat stress at the anthesis stage and post-anthesis stage, and the ABC transporters pathway was significantly enriched both by significantly changed metabolites and DEGs. There were 116 and 122 DEGs identified in the ABC transporter pathway in CK1 vs. HS1 and CK7 vs. HS7, respectively, all of which were annotated as ABC transporter genes (Appendix A). Based on annotation, these DEGs were classified into five subfamilies. The constitution was similar between the two stages, the subfamily B contained the highest number of ABC transporter DEGs followed by subfamilies G, C, A, and D (Appendix A). As reported different ABC transporter subfamily members play different roles, subfamilies B and G were the most active members in responses to hormones and abiotic stress [46], and we found that the subfamilies B and G were the most active ABC transporters responded to heat stress in wheat flag leaves during the reproductive stage.

ABCB transporters are primarily involved in the distribution and mediation of auxin inside the plant [47,48], which is a key regulator of plant reproductive development including anther and endosperm development, pollen fertility, and grain filling, and is highly sensitive to drought and heat stresses [49,50,51]. Heat stress significantly reduced the endogenous auxin content of rice panicles, and exogenous IAA application could reduce the membrane lipid peroxidation, ROS accumulation in spikelets, and the impact of heat on the pollen viability and yield component traits in rice [52]. AtABCB1 and AtABCB19 were reported as IAA efflux transporters, and the mutant plants exhibited dwarfism [53]. In this study, the *AtABCB1* and *AtABCB19* homologous genes (*TaABCB1*: *TraesCS7B02G181700*, *TraesCS7D02G284100*, *TraesCS7A02G284900*; *TaABCB19*: *TraesCS2A02G521400*, *TraesCS2B02G550700*, *TraesCS2D02G523100*) were all significantly down-regulated at two stages. This indicates that heat stress may have a negative effect on the IAA efflux of wheat flag leaf and, therefore, impact on wheat spike development during the reproductive stage.

One of the best-known roles of ABC transporters is the exportation of lipids from epidermal cells to the plant surface, which is essential for synthesizing waxy protective cuticle coats protecting plants from abiotic stress, especially under drought stress [12]. ABC transporter subfamily G is considered to participate in a wide range of biotic and abiotic stress responses of plants. In this study, a large number of DEGs encoding ABCG transporter were detected after heat stress. *AtABCG32* was strongly expressed in leaf, stem, and flower and crucial for the formation of a functional cuticle in *Arabidopsis* [54]. In this study, we detected a significantly increased expression of *TaABCG32* (*TraesCS6D02G166100* and *TraesCS3A02G217900*) in wheat flag leaves at the anthesis stage, which indicated that the up-regulated of these two ABCGs may facilitate the formation of waxy protective cuticle coats as a response under heat stress. In addition, many ABCGs were reported to be involved in forming a protective surface structure for reproductive cells to withstand changing environments [55,56,57]; however, the expression of homologous genes in wheat was insignificantly changed or undetected. This may be caused by these ABCGs having a specific expression in reproductive organs, as they functionally protect the plant spore or pollen.

ABC transporter C subfamily was reported to be involved in the detoxification of toxic compounds [58,59,60]. In this study, a large number of ABCCs were significantly changed under heat stress, which indicates that ABCCs may also participate in response to heat stress, but the specific role of ABCCs needs to be investigated.

In this study, the ABC transporter pathway was also significantly enriched by metabolites, most of which were amino acids. There was little information about any specific ABC transporters participating in amino acids transportation in plants; however, there was some evidence indicating ABC transporters can transport amino acids in microorganisms [61,62]. As the ABC transporter family was conserved among species of all kingdoms [12], we can believe that ABC transporter could transport amino acids in plants. This could facilitate plants to uptake nutrients including metals, amino acids, and peptides when they encounter stress conditions.

### 3.3. The Accumulation Pattern of Free Amino Acids Under Heat Stress

Accumulation of free amino acids after abiotic stress has been repeatedly reported in plants [14,15,16]. The increase in free amino acids can result from the degradation of stress-induced protein, and plants may also synthesize particular amino acids that play a specific beneficial role in stress response. Previous studies showed that alanine, valine, methionine, lysine, isoleucine, leucine, and phenylalanine were significantly increased by osmotic stress in *Arabidopsis thaliana* [23]. In line with a previous study, the content of valine, methionine, lysine, isoleucine, leucine, and phenylalanine was also increased after heat stress at two stages (Appendix A). Batista-Silva investigated the effect of drought stress on the amino acid metabolism of *Arabidopsis thaliana*, which revealed a broad range of amino acids were significantly increased, thirteen of which significantly increased after PEG treatment [63]. Among them, leucine, tyrosine, valine, isoleucine, tryptophan, proline, phenylalanine, lysine, glutamine, and methionine were also significantly increased after heat stress at two stages tested in this study (Figure 9A–J). These results indicated that some amino acids were able to participate in multiple stress responses. Arginine was reported as an alleviation of plant oxidative damage caused by abiotic stresses and can reduce the content of lipid peroxidation [64,65]. In this study, arginine was the most abundantly accumulated after heat stress, especially at the anthesis stage (Figure 9K). The transcriptional result indicated that the arginine biosynthesis pathway was activated under heat stress at the two stages, as most DEGs were up-regulated including genes coding acetylglutamate kinase, acetylornithine aminotransferase, and ornithine carbamoyltransferase (Appendix A). This indicated that wheat may activate arginine biosynthesis to relieve the oxidative stress caused by heat stress. Proline was well established as a compatible osmolyte and relevant for increased drought resistance acting as a buffer of the cellular redox status [66,67]. Proline was proved to confer heat tolerance in creeping bentgrass by increasing turf quality and leaf chlorophyll content. The proline application could induce most free amino acids under heat-stress condition [68]. In addition, studies showed that the content of proline is positively associated with stress tolerance [69,70]. In this study, the content of proline was significantly induced under heat stress at two stages, which may help the wheat release from ROS burst caused by heat stress. Based on the transcriptome result, the proline biosynthesis module (glutamate => proline) included by arginine and proline metabolism pathway was inhibited at the anthesis stage, all the DEGs encoding pyrroline carboxylate synthetase (*TraesCS1D02G280700*, *TraesCS3B02G395900*) in this module were down-regulated, and one DEG (*TraesCS1D02G212400*) encoding proline dehydrogenase and five DEGs (*TraesCS1B02G186400*, *TraesCS4D02G229700*, *TraesCS6A02G410600*, *TraesCS6B02G456400*, *TraesCS6D02G393500*) encoding prolylhydroxylase, which could degrade proline were up-regulated at anthesis stage (Appendix A). These results indicated that the increased proline after heat stress, at the anthesis stage, was caused by protein degradation rather than biosynthesis. Phenylalanine and tyrosine could be induced by heat stress in plants, which are flavonoid synthesis precursors, and it is reported that flavonoids are involved in a wide range of plant stress responses [71,72]. In this study, the increase in phenylalanine and tyrosine may facilitate flavonoid synthesis to participate in heat-stress response. The BCAAs were reported to accumulate in response to osmotic stress due to its increased biosynthesis [14,73]. In this study, all the BCAAs were significantly increased after heat stress, and the DEGs (*TraesCS4A02G059800*, *TraesCS4B02G235400*) that encode BCAA aminotransferase were significantly increased at both anthesis stage and post-anthesis stage, which are thought to be the key regulator of BCAAs and serves as the last step of BCAA biosynthesis. This indicated that BCAA biosynthesis may have a specific role in heat-stress response in wheat.

Free amino acids have been reported to participate in various abiotic stress in plants including osmotic stress, drought stress, salt stress, and oxidative stress, but the free amino acids change pattern caused by heat stress has not been reported in wheat. Therefore, we further investigated the amino acids profile in ten different wheat varieties both at normal conditions and heat-stress treatment during the post-anthesis stage (Appendix A), when heat stress caused more significantly changed metabolites and DEGs, and the average content of amino acids could reflect the overall change pattern caused by heat stress. All the significantly up-regulated amino acids detected in metabolism were also significantly up-regulated in the ten cultivars’ average content analysis except the proline, which did not reach the significant threshold (*p* < 0.05), and the changing pattern of each amino acid among most of the cultivars showed a consistent change pattern, which suggested a reliable amino acid responsive pattern to heat stress in wheat flag leaves. Furthermore, given that the amino acid response patterns are consistent across diverse wheat genotypes, the role of free amino acids in heat-stress responses may be applied to enhance the heat-stress resistance of other wheat cultivars.

Above all, the heat-stress response of wheat flag leaves during the reproductive stage was investigated; as a result, the genes that encode HSPs and ABC transporters, and amino acids were highlighted, which could not only provide potential metabolic or genetic markers for heat-resistant germplasms identification but also valuable targets for enhancing heat resistance through genetic manipulation.

## 4. Materials and Methods

### 4.1. Plant Materials and Growing Conditions

A semi-winter wheat cultivar “Cang 6005”, which has relatively high heat resistance, was cultivated in the wheat experimental field until it reached the reproductive stage. Twenty entire wheat plants, including the roots and the adhering soil, were removed from the experimental field at the anthesis stage (28 weeks) and 7 days post-anthesis (29 weeks), respectively. The 20 wheat plants were divided into two groups, with each group containing 10 plants, and transferred into two separate incubators for 1 h treatment. The temperature of one incubator was set at 25 °C as the control (CK1, CK7), and that of the other was set at 37 °C as heat stress (HS1, HS7). The flag leaves were collected, and three replicates were set for further experiments, with each replicate consisting of at least 10 flag leaves from three wheat plants.

### 4.2. RNA-Seq

RNA isolation: The flag leaf samples at the anthesis stage and post-anthesis stage were harvested after heat stress treatment and control. The wheat flag leaf total RNA was extracted using the RNAprep Pure Plant Kit (Tiangen, Beijing, China) according to the instructions provided by the manufacturer. The total RNA samples were qualified by agarose gels and Bioanalyzer 2100 system (Agilent Technologies, Santa Clara, CA, USA). Library preparation for transcriptome sequence: A total amount of 1 μg RNA per sample was used as input material for the RNA sample preparations. Sequencing libraries were generated using NEBNext^®^ Ultra^TM^ RNA Library Prep Kit for Illumina^®^ (NEB, Ipswich, MA, USA). NEBNext Adaptor with hairpin loop structure was ligated to prepare for hybridization. To select cDNA fragments of preferentially 250~300 bp in length, the library fragments were purified with the AMPure XP system (Beckman Coulter, Beverly, MA, USA). Then, 3 μL USER Enzyme (NEB, USA) was used with the size-selected sample. Then, PCR was performed with Phusion High-Fidelity DNA polymerase, Universal PCR Primers, and Index (X) Primer. At last, PCR products were purified (AMPure XP system) and library quality was assessed on the Agilent Bioanalyzer 2100. The clustering of the index-coded samples was performed on a cBot Cluster Generation System, and the library preparations were sequenced on an Illumina platform and 150 bp paired-end. The data quality was controlled and filtered by Fastp (v 0.19.3) to remove reads with adapters and the paired reads with low quality (Q ≤ 20). Mapping of reads: HISAT v2.1.0 was used to compare clean reads to the wheat reference genome (RefSeq v1.1). Gene expression levels and difference analysis: Feature Counts v1.6.2/String Tie v1.3.4d were used to calculate the gene alignment and FPKM. DESeq2 v1.22.1/edgeR v3.24.3 was used to analyze the differential expression between the comparison groups. The FDR and fold change were used as the threshold (fold changed ≥ 2 and FDR < 0.05) for the differential expressed gene. DEGs enrichment: The enrichment analysis was performed based on the hypergeometric test. For KEGG, the hypergeometric distribution test was performed with the unit of the pathway; for GO (gene ontology), it was performed based on the GO term. The significant threshold for KEGG and GO was set as *p*-value < 0.05.

### 4.3. Metabolism Analysis

Sample preparation and extraction: biological samples were freeze-dried by a vacuum freeze-dryer (Scientz-100F). A mixer mill (MM 400, Retsch, Haan, Germany) was used to crush the freeze-dried samples. Lyophilized powder (100 mg) was dissolved with 1.2 mL 70% methanol solution, and vortexed 30 s every 30 min for 6 times in total; the samples were placed in a refrigerator at 4 °C overnight, and centrifuged at 12,000 rpm for 10 min; and the supernatant was filtrated (SCAA-104, 0.22 μm pore size) for UPLC-MS/MS analysis. UPLC-MS/MS analysis: the sample extracts were analyzed by a UPLC-ESI-MS/MS system (UPLC, SHIMADZU Nexera X2; MS, Applied Biosystems 4500 Q TRAP, Carlsbad, CA, USA). The injection volume was 4 μL. The effluent was alternatively connected to an ESI-triple quadrupole-linear ion trap. LIT and triple quadrupole scans were acquired on a triple quadrupole-linear ion trap mass spectrometer, AB4500 Q TRAP UPLC/MS/MS System, equipped with an ESI Turbo Ion-Spray interface, operating in positive and negative ion mode and controlled by Analyst 1.6.3 software (AB Sciex, Framingham, MA, USA). Triple quadrupole scans were acquired as MRM experiments with collision gas (nitrogen) set to medium. DP and CE for individual MRM transitions were performed with further DP and CE optimization. A specific set of MRM transitions was monitored for each period according to the metabolites eluted within this period. The qualitative and quantitative metabolites were based on matching the retention time, count per second, and area of the peak to the Metware database (Metware Biotechnology, Wuhan, China) including more than 20,000 types of metabolites and secondary metabolites. Unsupervised PCA was performed using the Metware Cloud (https://cloud.metware.cn). Significantly changed metabolites were determined by fold change ≥ 1.5 or fold change ≤ 0.67, and VIP ≥ 1. Metabolites KEGG annotation were annotated based on KEGG compound database (http://www.kegg.jp/kegg/compound/, accessed on 26 January 2025), and *p*-value < 0.05 was set as the significant threshold.

### 4.4. Ten Wheat Cultivars’ Amino Acid Profile Analysis

Ten wheat cultivars (Sheng126, Shannong10-2, Jinmai24, Jinan13, Pingan7, Sheng125, Shannong23, 04zhong36, Huaimai26, Lian0809) with different genetic backgrounds and heat-resistant levels were selected to compare the change in amino acid profile caused by heat stress. Ten plants of each selected wheat cultivar were removed from the field at the post-anthesis stage to an incubator for 1 h, 37 °C heat stress treatment, or 25 °C for control, respectively. After treatment, thirty flag leaves were collected and smashed in liquid nitrogen. A total of 0.05 g of the flag leaves was mixed with 500 µL of 70% methanol/water and vortexed. A total of 300 μL of supernatant was taken into a new centrifuge tube and placed the supernatant in a −20 °C refrigerator for 30 min, Then, the supernatant was centrifuged and transferred 200 μL through a Protein Precipitation Plate for further analysis. The sample extracts were analyzed using an LC-ESI-MS/MS system (UPLC, ExionLC AD; MS, QTRAP^®^ 6500+System). ESI-MS/MS: AB 6500+QTRAP^®^ LC-MS/MS System, equipped with an ESI Turbo Ion-Spray interface, operated in both positive and negative ion modes and controlled by Analyst 1.6 software (AB Sciex). A specific set of MRM transitions was monitored for each period according to the amino acid eluted within this period. Two-tailed unpaired Student’s *t*-tests (*p*-value < 0.05) were used to assess the statistical significance of the differences in amino acids.

### 4.5. RT-qPCR Analysis

To confirm transcriptional changes in the response to heat stress, the same samples for the RNA-seq analysis were used for RT-qPCR analysis. Total RNA was isolated and cDNA was synthesized for each biological replicate using RNAprep pure Plant kit (Tiangen) and FastKing RT kit (Tiangen), according to the manufacturer’s instructions. RT-qPCR was performed on an ABI 7500 (Applied Biosystem) using TB Green Premix Ex Taq II (Takara Biomedical Technology, Beijing, China). The cDNA (100 ng) was used in a 10 μL reaction with three technical replicates. The thermocycler settings were as follows: 30 s at 95 °C, followed by 40 cycles of 5 s at 95 °C, 30 s at 60 °C, and 10 s at 72 °C. Melt curves were run for all primer pairs (Appendix A), and the *TaRP15* gene was used as a reference. Data were analyzed using the ABI 7500 software. Two-tailed unpaired Student’s t-tests (*p*-value < 0.05) were used to assess the statistical significance of the expression differences.

## 5. Conclusions

In this study, the rearrangement of heat stress in wheat flag leaves was investigated on transcriptional and metabolic levels during the reproductive stage. Some pathways were highlighted due to the enrichment of significantly changed metabolites and DEGs, including protein processing in endoplasmic reticulum, ABC transporters, and free amino acids. A group of DEGs encoding a wide range of molecular chaperones were activated by heat stress, especially the HSP20 family which responded to heat stress most intensely. In addition, ABC transporters were deeply involved in heat-stress response that many members of this family were significantly changed by heat stress and could be classified into five subfamilies, and most of them come from subfamilies B and G. The heat-stress response pattern of free amino acid was examined in this study, tyrosine, isoleucine, tryptophan, arginine, phenylalanine, methionine, leucine, valine, lysine, and glutamine could be induced by heat stress in different wheat cultivar’s flag leaves. Above all, this study gives some reliable clues for further heat-stress response mechanism studies on transcriptional and metabolic levels.

## Figures and Tables

**Figure 1 ijms-26-01468-f001:**
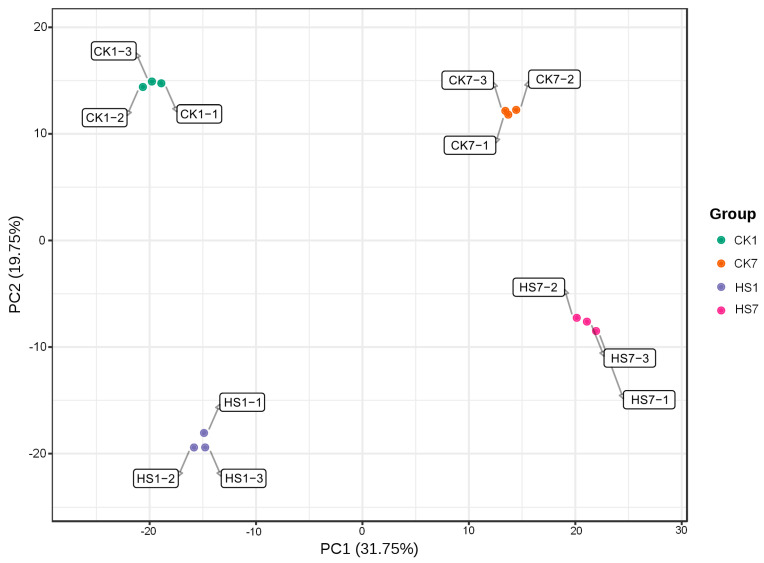
PCA analysis of wheat flag leaves harvested at anthesis and 7 days post-anthesis stage under control and heat stress treatment. Score plots are based on the metabolite abundances of four groups (CK1, HS1, CK7, and HS7). PCA reveals the distinct divergence of metabolite change patterns in response to heat stress at the anthesis and post-anthesis stages. The four rounded markers with different colors in each ellipse represent three biological replicates.

**Figure 2 ijms-26-01468-f002:**
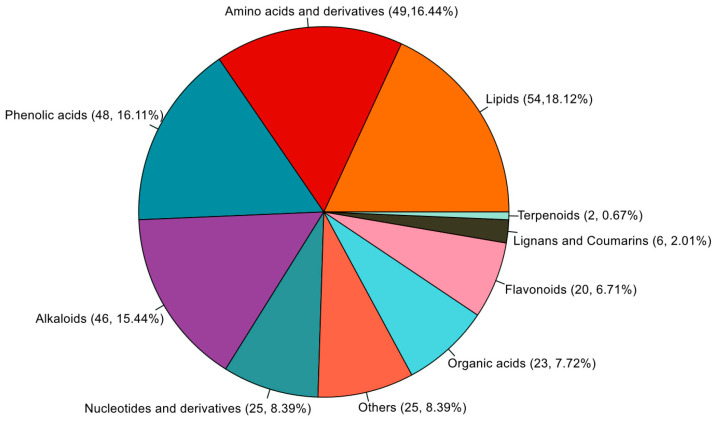
Percentage composition of various non-redundant metabolite types that significantly changed in the comparisons of CK1 vs. HS1 and CK7 vs. HS7.

**Figure 3 ijms-26-01468-f003:**
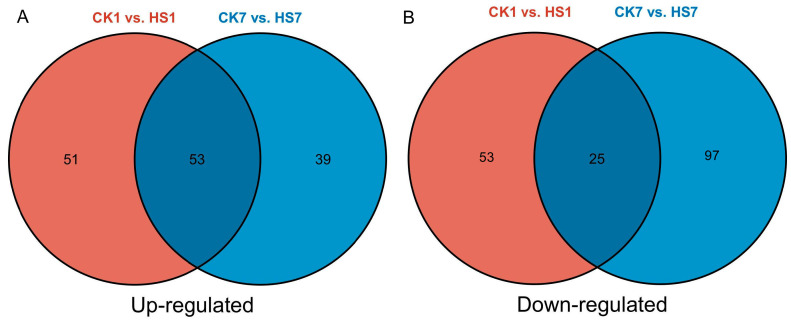
Venn diagram of significantly changed metabolites. Venn diagram of up-regulated metabolites (**A**). Venn diagram of down-regulated metabolites (**B**). The black numbers indicate the numbers of significantly changed metabolites.

**Figure 4 ijms-26-01468-f004:**
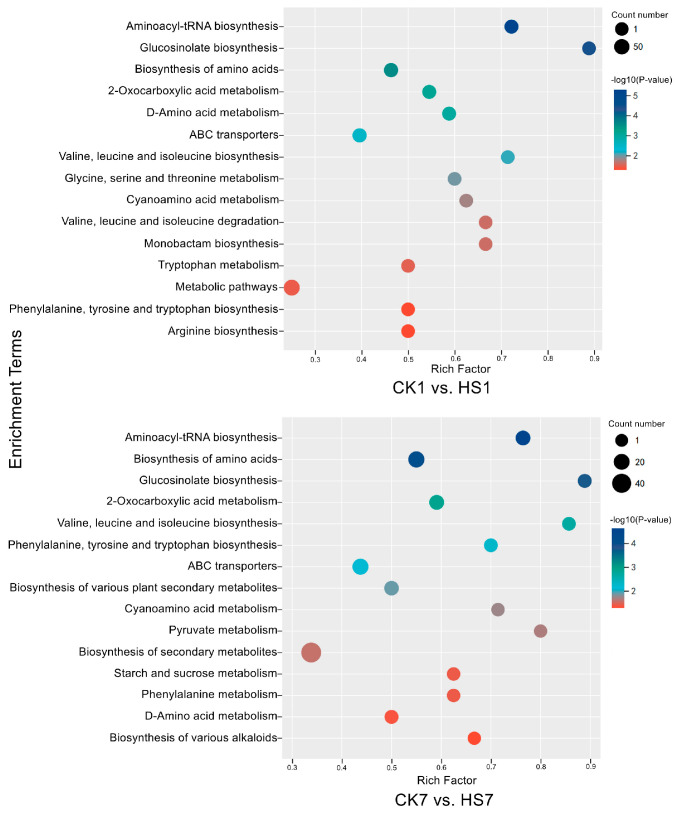
KEGG enrichment bubble diagrams of significantly changed metabolites in CK1 vs. HS1 and CK7 vs. HS7, respectively. Bubble size indicates the number of DEGs enriched in this KEGG pathway; bubble color indicates the *p*-values.

**Figure 5 ijms-26-01468-f005:**
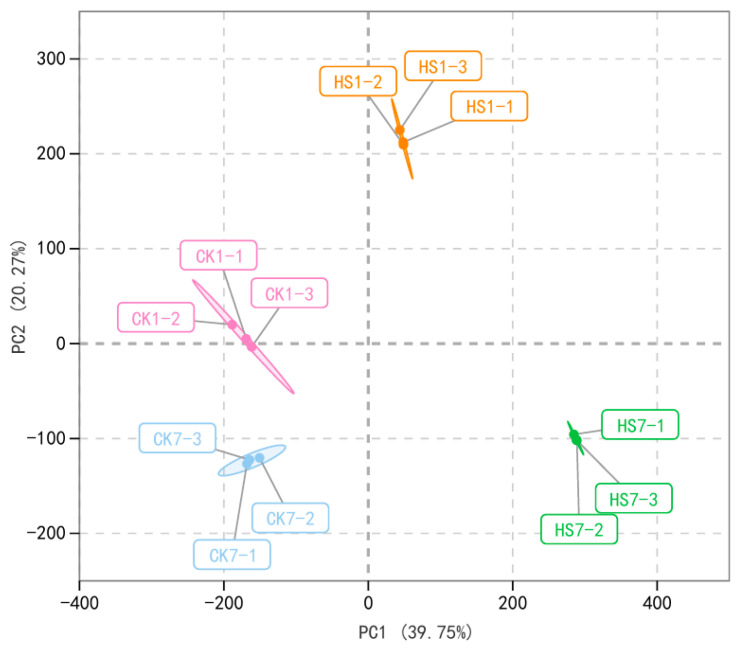
PCA analysis of wheat flag leaves harvested at anthesis and 7 days post-anthesis stage under control and heat stress treatment. Score plots are based on the transcript abundances of four groups (CK1, HS1, CK7, HS7). PCA reveals the distinct divergence of metabolite change patterns in response to heat stress at the anthesis and post-anthesis stages. The four rounded markers with different colors in each ellipse represent three biological replicates.

**Figure 6 ijms-26-01468-f006:**
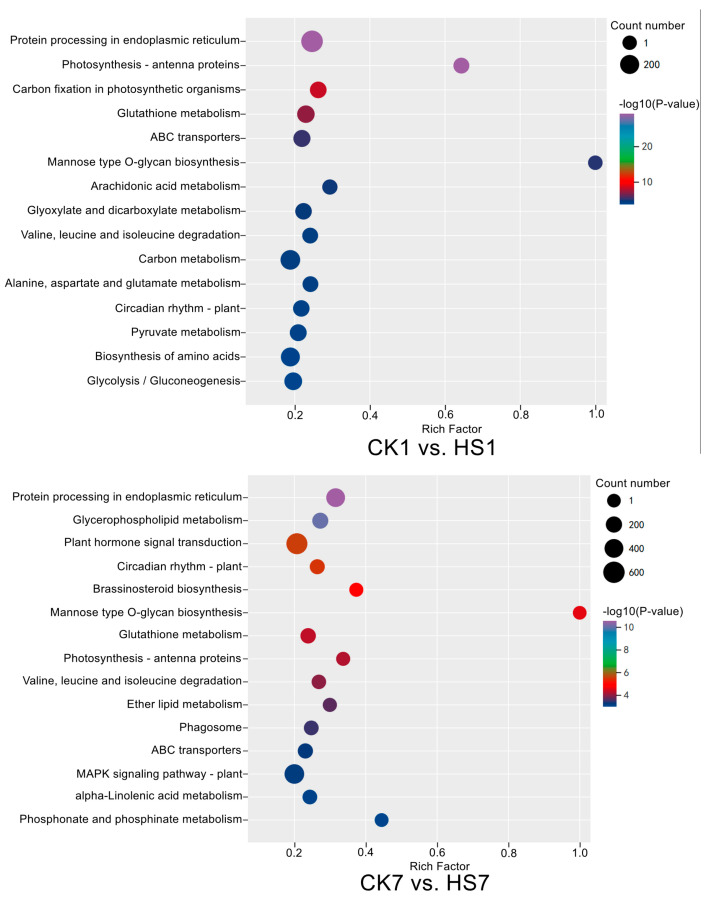
KEGG enrichment bubble diagrams of DEGs in CK1 vs. HS1 and CK7 vs. HS7, respectively. Bubble size indicates the number of DEGs enriched in this KEGG pathway; bubble color indicates the *p*-values.

**Figure 7 ijms-26-01468-f007:**
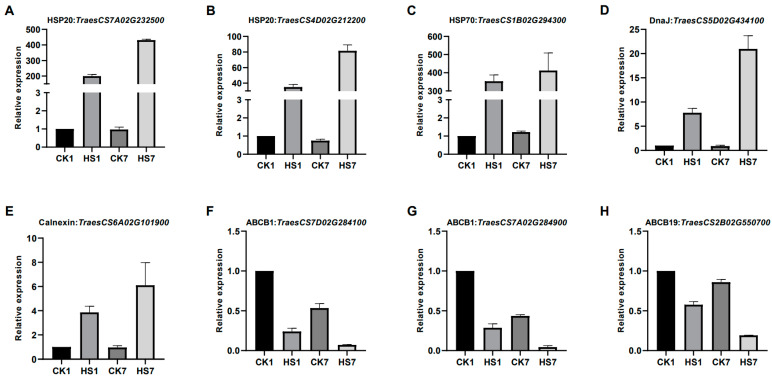
RT-qPCR verification of expression levels of selected genes identified by RNA sequencing. The relative expression levels were analyzed by RT-qPCR and calculated by 2^−ΔCt^. (**A**,**B**): heat shock protein 20, (**C**): heat shock protein 70, (**D**): DnaJ, (**E**): Calnexin, (**F**–**H**): ABC transporter.

**Figure 8 ijms-26-01468-f008:**
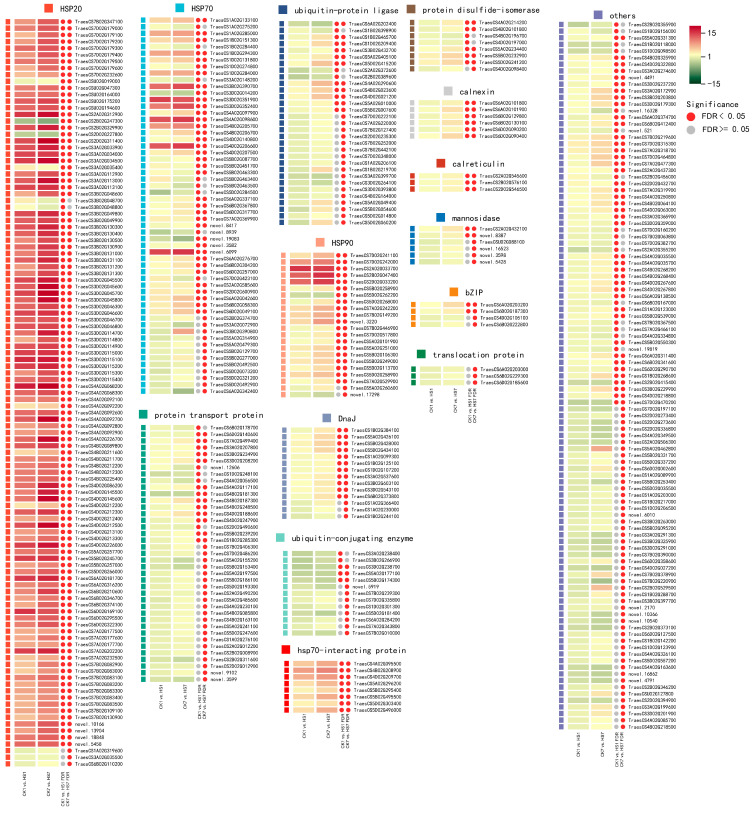
The heatmap of DEGs involved in the protein processing in endoplasmic reticulum pathway. The DEGs of two comparison groups are classified into 15 categories. The color of each vertical column represents the up (red) or down (green) fold change (log2) of FPKM; the red plot indicates the gene is significantly changed after heat stress treatment.

**Figure 9 ijms-26-01468-f009:**
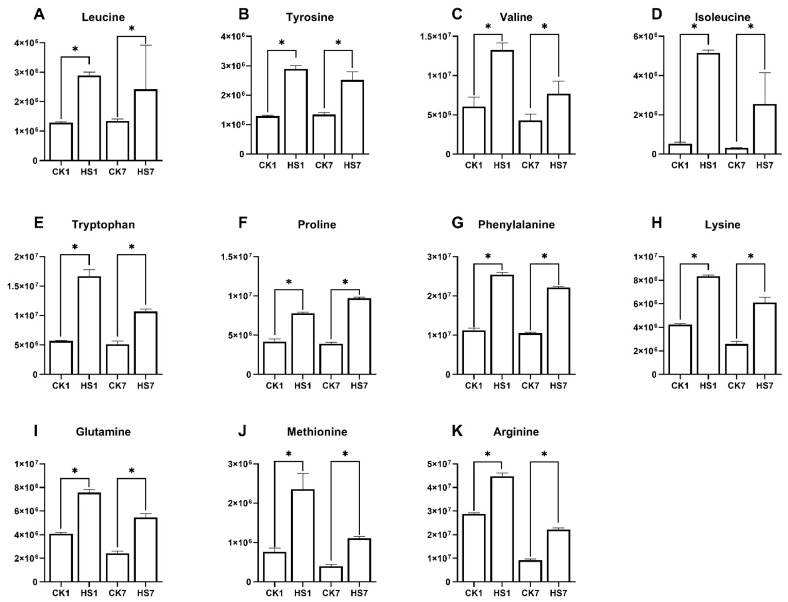
The content of 11 amino acids exposed to heat stress at the anthesis stage and post-anthesis stage. The bar graphs present the results of the metabolism data and the means ± SD of three biological replicates; *, *p* < 0.05; (**A**): leucine, (**B**): tyrosine, (**C**): valine, (**D**): lsoleucine, (**E**): tryptophan, (**F**): proline, (**G**): phenylalanine, (**H**): lysine, (**I**): glutamine, (**J**): methionine, (**K**): arginine.

## Data Availability

Data are contained within the article and Appendix A.

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
