# Peer review of "The Effect of Heat Stress on Wheat Flag Leaves Revealed by Metabolome and Transcriptome Analyses During the Reproductive Stage"

_ijms, 2025, doi:10.3390/ijms26041468_

Round 1

Reviewer 1 Report

Comments and Suggestions for Authors

This study investigates the impact of heat stress on wheat flag leaves during the reproductive phase, utilizing metabolome and transcriptome analyses. The research aims to understand how heat stress affects wheat flag leaves by identifying significant changes in metabolites and differentially expressed genes (DEGs) during the anthesis and post-anthesis stages. The methodology adopted by the authors includes Metabolome Analysis (using UPLC-MS/MS to detect and quantify metabolites in wheat flag leaves at different developmental stages and under heat stress conditions), Transcriptome Analysis (using RNA-seq to identify DEGs in response to heat stress), and Data Integration (integrating metabolome and transcriptome data to provide a comprehensive view of heat stress responses).

The study concluded that the post-anthesis stage is more sensitive to heat stress, with a higher number of differentially expressed metabolites and genes. Amino acids and their derivatives play a significant role in the heat stress response, with several amino acids being significantly regulated. Heat shock proteins (HSPs), especially the HSP20 family, are strongly activated by heat stress. ABC transporters play a crucial role in the heat stress response, with specific subfamilies being more active. In this sense, the study provides insights into the heat stress response in wheat flag leaves, identifying key metabolites and genes involved. The findings offer potential targets for improving heat stress tolerance in wheat.

The literature review is comprehensive, well-structured, and provides a solid context for the study, demonstrating a deep understanding of the subject and justifying the importance of the research conducted. The methodology is well described and appropriate for the study's objectives, including advanced techniques in metabolomic and transcriptomic analysis, use of controls, biological replicates, and validation of results by RT-qPCR. The results are presented logically and coherently, with effective use of figures and tables to illustrate the findings. The comparative analysis between different conditions and developmental stages is clear and detailed. The discussion integrates the results well with previous studies, highlighting the importance of the findings and providing a solid interpretation of the data. The conclusions effectively summarize the main findings of the study, highlighting the areas of greatest impact and relevance.

The study addresses a relevant topic, especially in the context of climate change and the need to develop wheat cultivars more resistant to heat stress. The work provides new information on the mechanisms of heat stress response in wheat, which can contribute to future research and genetic improvement.

Congratulations to the authors for their work!

The manuscript under review seems to me to be of very high quality, justifying its publication. This was clear in the review sent previously. Even so, here are some notes that will help improve the work.
The manuscript entitled ‘The effect of heat stress on wheat flag leaves revealed by metabolome and transcriptome analyses during the reproductive stage’ investigates the impact of heat stress on wheat flag leaves using metabolome and transcriptome analyses the study identifies significant changes in metabolites and differentially expressed genes (DEGs) in the anthesis and post-anthesis phases highlighting the sensitivity of heat stress to the reproductive stage the main findings include the enrichment of aminoacyl-tRNA biosynthesis pathways and the up-regulation of heat shock proteins (HSPs). The main findings include the enrichment of aminoacyl-tRNA biosynthesis pathways and the up-regulation of heat shock proteins (HSPs) and ABC transporters.
Some aspects that could be improved:
1) Lack of detailed information on the duration and exact conditions of the heat stress treatment and the need for a more comprehensive description of the experimental setup the control conditions are also not described in sufficient detail being important to specify the temperature and duration of the control treatment to ensure a clear comparison with the heat stress conditions.
2) Regarding data analysis the manuscript does not provide sufficient information on the statistical methods used to analyse the data including details on the statistical tests the significance thresholds and any corrections for multiple comparisons the number of biological replicates and techniques used in the study should be clearly stated as this information is crucial to assess the reliability of the results.
3) In the interpretation of the results the biological relevance of the findings is not discussed in depth providing more context on how these changes impact wheat physiology and stress tolerance would increase the significance of the manuscript.
4) Minor problems include the legibility of some figures and tables which are difficult to read due to small font sizes and clustered data points improving the clarity and presentation of these visual elements would help with the interpretation of the results figure and table legends should be more descriptive providing enough information to understand the data without having to refer to the main text.
5) The clarity of the text could also be improved with additional explanations or simplifications in certain sections of the manuscript particularly those describing complex biochemical pathways to make them more accessible to a wider audience there are instances of redundant text that could be simplified to improve the overall flow and readability of the manuscript.
6) References should be checked to ensure that all statements especially those related to previous studies and established knowledge are properly cited this will increase the credibility of the manuscript and provide readers with sources for further reading.
In conclusion the manuscript presents valuable insights into the molecular mechanisms of the heat stress response in wheat flag leaves addressing the major and minor problems described above will significantly improve the clarity, reproducibility and impact of the study.

Reviewer 2 Report

Comments and Suggestions for Authors

This manuscript is well written, and my comments are as follows.

1. Page 1, line 23: I realize that DEG stands for differential expressed gene. Don’t use abbreviation in the abstract.

2. Page 7, Fig 4: Pic sizes are too small. Please consider to enlarge it as Fig 6 (page 9).

3.Page 9, Fig 6: Pic sizes are too big. If the authors resize them smaller, the figure legend at next page can be place on the same page.

4. Page 13, Figure 8: Is it possible to enhance resolution of the figure, so that readers can see them.

Reviewer 3 Report

Comments and Suggestions for Authors

The paper entitled: “The effect of heat stress on wheat flag leaves revealed by metabolome and transcriptome analyses during the reproductive stage” by Duan et al. discusses the effect of heat stress on the metabolic pathway of wheat plants. The manuscript is well-written, containing promising data suitable for publication in IJMS after major revision. Here some issues should be addressed before being accepted for publication.

1-     The introduction should be concise and targeted.

2-     The clear hypothesis and aim of the study should be clarified at the end of the introduction.

3-     The authors should be giving more clarification for why specific metabolite groups such as terpenoids are less affected by heat stress compared to other metabolites.

4-     The authors should clarify how the obtained results contribute to improving the tolerance of wheat to heat stress.

5-     The authors should be given more information about metaware database for the validation process to ensure the reliability of the results.

6-     The authors should clarify whether the results obtained are suitable for different wheat verities or different environmental conditions

7-     Although the flavonoids are the largest plant metabolite group but it’s the lowest one affected by heat stress, why?

8-     Although Fig. 1 is informative, it lacks some important data that increases the clarity of the results. For instance, the figure lacks a percentage of variance that explains each component.

9-     The authors should link between upregulation of some metabolites such as alkaloids and amino acids with heat stress mechanisms (cell signaling, osmoprotection, etc..).

10- The authors should integrate KEGG with GO analysis to highlight the cellular components, biological processes, and molecular functions that are affected by heat stress.

11- The growing conditions and heat stress should be discussed in detail.

12- The authors should clarify the age of plants at cthe ollected sample in the anthesis stage and post-anthesis stage

Comments on the Quality of English Language

The manuscript need moderate English editing

Reviewer 4 Report

Comments and Suggestions for Authors

The manuscript "The effect of heat stress on wheat flag leaves revealed by metabolome and transcriptome analyses during the reproductive stage" is of exceptional quality and comprehensive. At a time of significant global climate disruption, heat stress in one of the most important cereals is worth researching, as it directly affects yield.

In my opinion, more information should be provided about the wheat cultivar, as the conclusion drawn may not apply equally to all types of wheat. In the introduction, it mentions diploid species... This part of the text should be expanded, with additional details about the species used in the research.

Round 2

Reviewer 3 Report

Comments and Suggestions for Authors

The manuscript is suitable for publication in the current form